# Similar Characteristics of siRNAs of Plant Viruses Which Replicate in Plant and Fungal Hosts

**DOI:** 10.3390/biology11111672

**Published:** 2022-11-17

**Authors:** Tianxing Pang, Jianping Peng, Ruiling Bian, Yu Liu, Dong Zhang, Ida Bagus Andika, Liying Sun

**Affiliations:** 1State Key Laboratory of Crop Stress Biology for Arid Areas, College of Plant Protection, Northwest A&F University, Xianyang 712100, China; 2State Key Laboratory of Crop Stress Biology for Arid Areas, College of Horticulture, Northwest A&F University, Xianyang 712100, China; 3College of Plant Health and Medicine, Qingdao Agricultural University, Qingdao 266109, China; 4Institute of Plant Science and Resources, Okayama University, Kurashiki 710-0046, Japan

**Keywords:** plant viruses, fungi, mycoviruses, RNA silencing, small interfering RNAs

## Abstract

**Simple Summary:**

RNA silencing in fungi was shown to confer antiviral defense against plant viruses. In this study, using high-throughput sequencing and bioinformatic analyses, we showed that small interfering RNAs (siRNAs) of cucumber mosaic virus and tobacco mosaic virus (TMV) which replicated in phytopathogenic fungi *Rhizoctonia solani* and *Fusarium graminearum* had similarities with viral siRNAs produced in plant hosts in regard to the size distributions, proportion of plus and minus senses, and nucleotide preference for the 5′ termini. Additionally, our results also determined that both *F. graminearum* DCL1 and DCL2 were involved in the production of TMV siRNAs. Thus, the fungal RNA silencing machineries have adaptive capabilities to recognize and process the genome of invading plant viruses.

**Abstract:**

RNA silencing is a host innate antiviral mechanism which acts via the synthesis of viral-derived small interfering RNAs (vsiRNAs). We have previously reported the infection of phytopathogenic fungi by plant viruses such as cucumber mosaic virus (CMV) and tobacco mosaic virus (TMV). Furthermore, fungal RNA silencing was shown to suppress plant virus accumulation, but the characteristics of plant vsiRNAs associated with the antiviral response in this nonconventional host remain unknown. Using high-throughput sequencing, we characterized vsiRNA profiles in two plant RNA virus–fungal host pathosystems: CMV infection in phytopathogenic fungus *Rhizoctonia solani* and TMV infection in phytopathogenic fungus *Fusarium graminearum*. The relative abundances of CMV and TMV siRNAs in the respective fungal hosts were much lower than those in the respective experimental plant hosts, *Nicotiana benthamiana* and *Nicotiana tabacum*. However, CMV and TMV siRNAs in fungi had similar characteristics to those in plants, particularly in their size distributions, proportion of plus and minus senses, and nucleotide preference for the 5′ termini of vsiRNAs. The abundance of TMV siRNAs largely decreased in *F. graminearum* mutants with a deletion in either *dicer-like 1* (*dcl1*) or *dcl2* genes which encode key proteins for the production of siRNAs and antiviral responses. However, deletion of both *dcl1* and *dcl2* restored TMV siRNA accumulation in *F. graminearum*, indicating the production of *dcl*-independent siRNAs with no antiviral function in the absence of the *dcl1* and *dcl2* genes. Our results suggest that fungal RNA silencing recognizes and processes the invading plant RNA virus genome in a similar way as in plants.

## 1. Introduction

RNA silencing, also termed RNA interference (RNAi), is a cellular gene downregulation mechanism mediated by 20–30-nucleotide-long small RNAs (sRNAs) [1,2]. RNA silencing is involved in various cellular processes and operates at transcriptional or post-transcriptional levels, with its core mechanism conserved among diverse eukaryotes [3,4,5,6]. Generally, RNA silencing is initiated by the processing of highly base-paired or double-stranded RNAs (dsRNAs), which are in many cases produced by cellular or viral RNA-dependent RNA polymerase (RdRP), into small interfering RNAs (siRNAs) or microRNAs (miRNAs) by the ribonuclease III-like enzymes Dicer or Dicer-like (DCL). These sRNAs then associate with RNA-induced silencing complexes (RISCs) containing Argonaute (AGO) proteins to mediate sequence-specific gene repression activities through suppression of gene/DNA transcription, RNA degradation, and inhibition of RNA translation [7,8,9]. Along with other regulatory functions, RNA silencing has been highly implicated in antiviral defense in various organisms, including mammals [10], insects [11], nematodes [12], plants [13], and fungi [14], as well as in nonviral plant immunity through trans-kingdom transfer of siRNAs [15,16].

As in other organisms, diverse RNA silencing-related pathways involved in the maintenance of genome stability and gene regulation have been uncovered in fungi [17,18,19,20], although some fungi were found to be deficient in RNA silencing mechanisms due to loss of the core genes in the RNA silencing pathway [21,22,23,24]. Like plants, fungi encode multiple homologs for each RdRP, AGO, and DCL protein, with a functional specialization for particular RNA silencing pathways [20,25]. Pioneering works on the ascomycetous *Neurospora crassa* have revealed a post-transcriptional gene-silencing phenomenon termed quelling, which is associated with multiple copies of integrated transgenes [26,27] and maintenance mechanisms of fungal genome stability during vegetative growth [28]. Subsequently, a similar quelling-like phenomenon was observed in fungi such as *Cryptococcus neoformans* and *Mucor circinelloides* [29,30]. Furthermore, studies on *N. crassa* have uncovered the presence of meiotic silencing by unpaired DNA (MSUD), an RNAi-related mechanism in which the fungus protects its genomic integrity during meiosis through the silencing of sequences that are unpaired during meiosis [31]. In *Schizosaccharomyces pombe*, RNA silencing machinery is required for heterochromatin formation through the involvement of RNA-induced transcriptional silencing complex (RITS) [32,33]. Aside from siRNAs that are associated with quelling, MSUD, and RITS, other classes of endogenous sRNAs which may have regulatory functions, including DCL-independent siRNAs, were also identified in fungi [3,17,34].

Like other eukaryotes, fungi naturally host viruses, commonly called mycoviruses [35]. Diverse mycoviruses have been identified in fungi, spanning at least 16 families and 24 genera of viruses [36,37]. The majority are viruses with RNA genomes (single- and double-stranded RNAs), whereas a much smaller number have circular single-stranded DNA genomes. Although most mycoviruses are asymptomatic to their fungal hosts, increasing numbers of mycoviruses have been found to alter the growth, morphology, and pathogenicity of the hosts [38,39]. Studies using *N. crassa* and some plant pathogenic ascomycetous fungi defective in genes encoding key RNA silencing proteins provide an understanding of the role of RNA silencing in antiviral defense in fungi. Ascomycetous fungi are generally known to encode two *dcl* genes, *dcl1* and *dcl2* [25]. Deletions of either one or both *dcl* genes in *Cryphonectria parasitica*, *Colletotrichum higginsianum, Fusarium graminearum*, *Sclerotinia sclerotiorum*, and *N. crassa* resulted in increases in mycovirus accumulation and/or symptom severity [14,40,41,42,43]. Likewise, deletions of one or two *Ago* genes disrupted the ability of fungi to defend against mycovirus infection [40,44,45]. Furthermore, numerous studies on diverse mycoviruses and fungal hosts have revealed the close association of mycovirus infection with the accumulation of vsiRNAs [40,41,43,46,47,48,49,50,51,52,53,54,55,56,57,58], further supporting the view that mycoviruses are generally targeted by RNA silencing-mediated defense in fungi.

Almost two decades ago, replication of a plant virus in *Saccharomyces* cerevisiae, a unicellular fungus (yeast), was demonstrated through artificial inoculation. *S*. *cerevisiae* has since been developed as a model host to study the mechanism of plant virus replication [59,60]. More recently, reports have shown that certain plant pathogenic filamentous fungi and oomycetes are suitable hosts of plant viruses. Through artificial inoculation, tobacco mosaic virus (TMV, genus *Tobamovirus*), among other plant RNA viruses, was shown to be capable of replicating in *Colletotrichum acutatum* and *Phytophthora infestans* (oomycete) [61,62]. TMV was also shown to replicate in *F. graminearum* by inoculation through the transfection of fungal protoplasts [63]. Moreover, *C. parasitica*, *Valsa mali*, *F. graminearum*, and the oomycete *Phytoptora infestans* can also host plant viroids, the subviral agents [64,65]. Importantly, our study previously discovered a natural infection of cucumber mosaic virus (CMV, genus *Cucumovirus*) in *Rhizoctonia solani* strains isolated from a potato plant [66]. Following this study, we recently also showed that by the screening of fungal strains isolated from virus-infected plants, various plant viruses were found to be commonly acquired by fungi [67]. These findings indicate that plant viruses can be transferred from plant to fungus during fungal colonization of the plant. TMV infection in *C. acutatum* induced gene silencing (virus-induced gene silencing) and is associated with the accumulation of vsiRNAs [61]. Deletion of both *dcl1* and *dcl2* genes in *F. graminearum* drastically elevated accumulations of TMV and hop stunt viroid [63,64]. Together, these observations indicate that fungal antiviral RNA silencing can recognize and target the invading plant viruses including viroids, but the characteristics of vsiRNAs associated with antiviral responses against plant virus infection in fungi have not been studied in detail.

In this study, we used deep sequencing analysis to characterize siRNAs derived from CMV and TMV that replicated in *R. solani* and *F. graminearum*, respectively. In addition, *F. graminearum* mutants defective in *dcl* genes were also included in the analyses. Our results revealed that the features of CMV and TMV siRNAs accumulated in fungi were similar to those accumulated in plants and confirmed the role of *F. graminearum* DCL1 and DCL2 in the biosynthesis of vsiRNAs with effective antiviral function.

## 2. Materials and Methods

### 2.1. Fungal Strains

*R. solani* strain Ra1 infected with CMV Rs isolates and CMV cured, and *R. solani* strain infected with CMV Fny isolate, have been described previously [66]. *Fusarium graminearum* (PH-1 strain) wild type, Δ*dcl1*, Δ*dcl2,* and Δ*dcl1*-Δ*dcl2* mutants infected with TMV have been described previously [63]. *R. solani* strain 80 (*R. solani* 80) was isolated from one of the potato tubers showing black scurf disease collected from potato-planting areas located in the middle and western parts of Inner Mongolia Province of China from the years 2008–2015 [66]. The presence of a virus in *R. solani* 80 was identified through high-throughput sequencing carried out in the previous study [66].

### 2.2. Plant Virus Inoculation

To inoculate plant viruses, CMV- and TMV-infected leaves were homogenized in 0.1 M phosphate buffer (pH 7.0) and rubbed onto carborundum-dusted leaves of *N. benthamiana* and *N. tabacum* plants, respectively. Noninoculated upper leaves were sampled seven days after inoculation.

### 2.3. Total RNA Extractions

Fungal total RNA was extracted from the mycelia of five-day-old PDB culture as described previously [63]. Briefly, fungal mycelia were homogenized in a buffer containing 100 mM Tris/HCl (pH 8.0), 200 mM NaCl, 4 mM EDTA, and 4% SDS, followed by phenol/chloroform extraction. After extraction, total RNA was treated with RQ1 DNase I to remove fungal DNA. Plant total RNA was extracted from leaves using TRIzol (Invitrogen, Waltham, MA, USA).

### 2.4. sRNA Sequencing 

The cDNA libraries of sRNA were prepared using Truseq SRNA Sample Preparation Kit (Illumina, San Diego, CA, USA) according to the supplied protocol. The sRNA cDNA library was amplified with TruSeq PE Cluster Kit (Illumina, San Diego, CA, USA) and then used for sequencing on an Illumina HiSeq2500 at Hanyu Bio-Tech (Shanghai, China). 

### 2.5. Bioinformatics Analysis

Adaptor sequences and poor-quality reads were removed by FASTX-toolkit (http://hannonlab.cshl.edu/fastx_toolkit/, Version 0.0.14, accessed on 15 September 2021). Reads which were less than 18 nt or more than 30 nt were removed using the Cutadapt software (https://cutadapt.readthedocs.io/en/stable/, Version 4.1, accessed on 15 September 2021). The remaining reads with lengths of 18–30 nucleotides were mapped using Bowtie software (http://bowtie-bio.sourceforge.net, Version 1.3.1, accessed on 15 September 2021) to the genome of TMV (NC_001367.1) with 1 mismatch allowed, CMV isolate Rs (RNA1 (MG025947), RNA2 (MG025948), and RNA3 (MG025949)), CMV isolate Fny (RNA1 (D00356), RNA2 (NC_002035) and RNA3 (D10538)) and Endorna-like virus (OP763640) with 0 mismatches allowed. The number of vsiRNA reads was normalized to “reads per million” (RPM) according to the number of total reads of the corresponding sRNA library. Homemade Perl scripts and Excel were used to analyze the 5′-terminal nucleotide and the distribution of vsiRNA in the genome. The ggplot2 package in R (Version 4.1.1) was used to facilitate making the plot of siRNA mapping. Raw data of sRNA libraries have been deposited to the SRA database with the accession number PRJNA900007.

## 3. Results

### 3.1. Characteristics of CMV- and Mycovirus-Derived siRNAs in R. solani

In a previous study, we discovered CMV infection in an *R. solani* strain (designated as Ra1 strain) isolated from a potato plant collected from a field in the Inner Mongolia Province of China [66]. This fungi-infecting CMV (referred to as Rs isolate) had a typical three-segmented RNA genome (Figure 1A) and showed the closest identity (>98%) to East Asian CMV isolates belonging to the subgroup Ia. CMV-cured Ra1 strains were obtained through single spore isolation. In that study, we also artificially obtained a CMV-infected *R. solani* strain by transfection of protoplasts of a virus-free *R. solani* strain with CMV Fny isolate (subgroup I CMV isolate). 

To characterize CMV siRNAs in *R. solani*, total RNAs were extracted from *R. solani* Ra1 and *R. solani* Ra1/CMV-cured (two biological replicate samples) and *R. solani*/CMV Fny and then subjected to high-throughput sequencing of sRNAs followed by bioinformatic analyses to investigate the sequence characteristics of vsiRNAs. Total RNA extracted from *N. benthamiana* infected with CMV Fny was also included in the high-throughput sequencing. In addition, to characterize mycovirus siRNAs in *R. solani*, high-throughput sequencing was also performed on total RNA extracted from an *R. solani* strain (*R. solani* 80) infected with a novel mycovirus related to endornaviruses (Appendix A), tentatively named Rhizoctonia solani endorna-like virus 1 (RsEnLV1, Figure 1A). Sequence analysis yielded around 10–25 million sRNA reads ranging from 18 to 30 nucleotides in size for each sRNA library (Table 1). Mapping of sRNA reads to the CMV genome revealed that high numbers of sRNAs were derived from CMV sequences in CMV-infected samples (*R. solani* Ra1 and *R. solani*/CMV Fny libraries) but no, or only very few, sRNAs were mapped to the CMV genome in *R. solani* Ra1/CMV-cured libraries (Table 1). This result suggests that CMV infection in *R. solani* is associated with the accumulation of vsiRNAs. The proportion of CMV siRNAs was higher in the *R. solani*/CMV Fny library (0.07%) than in the *R. solani* Ra1 libraries (0.03% and 0.02%), but both were much lower than that in the *N. benthamiana*/CMV Fny library, which was around half (53.03%) of the total sRNA reads (Table 1 and Figure 1B). The proportion of positive-strand (+) CMV siRNAs was similarly higher than negative-strand (−) siRNAs in all libraries (Figure 1B). Notably, much larger portions of CMV siRNAs in fungal samples were derived from RNA3 than, in order, RNA2 and RNA1. These differences were more pronounced in *R. solani* Ra1 than in the *R. solani*/CMV Fny library, whereas *N. benthamiana*/CMV Fny samples had similar proportions of RNA3- and RNA2-derived CMV siRNAs (Figure 1C). It is still unclear whether the higher proportion of CMV RNA3-derived siRNAs in the fungal host is related to the higher accumulation of RNA3 in fungi or due to other reasons. High numbers of sRNA reads in the *R. solani* 80 libraries were also mapped to RsEnLV1 genome sequence, whereas a much lower number of sRNA reads in the *R. solani* 80/virus-cured libraries were mapped to RsEnLV1 genome sequence (Table 1 and Figure 1D), suggesting that RsEnLV1 infection in *R. solani* is associated with accumulation of vsiRNAs.

CMV siRNAs in the *R. solani* Ra1 and *R. solani*/CMV Fny libraries were predominantly 21 and 22 nucleotides long, with peaks at 21 nucleotides, similar to the characteristics of CMV siRNAs in the *N. benthamiana*/CMV Fny library (Figure 2A). Differently, RsEnLV1 siRNAs in the *R. solani* 80 libraries had a broader size distribution; they were predominantly 21–24 nucleotides long, with peaks at 23 nucleotides (Figure 2B).

The 5′-terminal nucleotides of CMV siRNAs in the *R. solani* Ra1, *R. solani*/CMV Fny, and *N. benthamiana*/CMV Fny libraries had similar preferences; they were most biased toward C followed sequentially by U, A, and G (Figure 2C). In the *R. solani* 80 libraries, the 5′-terminal nucleotide of RsIM-as EnV1 siRNAs was most frequently U (Figure 2D). Thus, along with differences in size distribution, the 5′-terminal nucleotide preferences of siRNAs between CMV and mycoviruses in *R. solani* are also different.

CMV 21 and 22 nt siRNAs were distributed along the (+) and (−) strands of the CMV RNA1, 2, and 3 genomes, with several prominent siRNA hotspots, most of which were located at the same genome positions across the *R. solani* Ra1 and *R. solani*/CMV Fny libraries, while the distribution and hotspots of CMV siRNAs from the *N. benthamiana*/CMV Fny libraries along the (+) and (−) strands of the genomes were much more dense, due to the high number of siRNAs (Figure 3). Notably, in the fungal libraries, there was a prominent siRNA hotspot in the middle of the (+) strand of the CMV RNA3 genome, located at nucleotide position 950–971, which is the beginning of intercistronic region, but upstream of the initiation site of CMV CP subgenomic RNA synthesis [68]. Thus, the occurrence of this siRNA hotspot is not likely associated with CP subgenomic RNA accumulation. Similarly, RsEnLV1 siRNAs were also distributed along the (+) and (−) strands of virus genomes with several prominent siRNA hotspots (Appendix A). 

### 3.2. Characteristics of TMV siRNAs in Wild-Type and Dcl Mutants of F. graminearum

In a previous study, we reported that *F. graminearum* is a compatible host for TMV replication [63]. TMV was introduced by protoplast transfections to *F. graminearum* strain PH-1 wild-type, single *dcl* knockout mutants (Δ*dcl1* and Δ*dcl2*), and a double *dcl* knockout mutant (Δ*dcl1-*Δ*dcl2*) [63]. To characterize TMV siRNAs in *F. graminearum*, total RNAs were extracted from each TMV-infected wild-type and *dcl* knockout mutant strain, including the virus-free (VF) fungus, and then subjected to high-throughput sequencing of sRNAs. Additionally, total RNA samples extracted from *N. tabacum* infected with TMV were also included for sequence analysis. Two biological replicate samples were collected from each TMV–host combination and analyzed independently.

Sequence analysis yielded around 7–15 million sRNA reads ranging from 18 to 30 nucleotides in size for each sRNA library (Table 2). Mapping of sRNA reads to the TMV genome revealed that no sRNA in the *F. graminearum* wild-type VF sRNA libraries was mapped to the TMV genome, while considerable numbers of sRNAs in the TMV-infected *F. graminearum* wild-type sRNA libraries (0.02% and 0.04%) were derived from TMV genome sequences, albeit far fewer than those mapped to the TMV genome in the TMV-infected *N. tabacum* sRNA libraries (27.33% and 10.74%) (Table 2 and Figure 1E). Notably, there were far fewer TMV siRNAs in either *F. graminearum* single Δ*dcl1* or Δ*dcl2* mutant sRNA libraries than in the wild-type samples; surprisingly, however, TMV siRNAs were abundant in the double Δ*dcl1*-Δ*dcl2* mutant sRNA libraries, to an even greater extent than in wild-type sRNA libraries (Table 2 and Figure 1E). This result shows that deletion of either Δ*dcl1* or Δ*dcl2* drastically reduces TMV siRNA accumulation, but in the absence of both Δ*dcl1* and Δ*dcl2*, TMV-derived sRNAs are produced through a DCL-independent pathway.

TMV siRNAs in the *F. graminearum* wild-type and all Δ*dcl* mutant libraries were predominantly 21 and 22 nucleotides long, with peaks at 21 nucleotides (Figure 4A), similar to the size distribution of those in TMV-infected *N. tabacum* libraries (Figure 4B). The 5′-terminal nucleotide of TMV siRNAs in the *F. graminearum* wild-type and all Δ*dcl* mutant libraries as well as in TMV-infected *N. tabacum* libraries had a similar preference, with U the most frequent, followed sequentially by A, C, and G (Figure 4C). Interestingly, TMV siRNAs between the *F. graminearum* wild-type and double Δ*dcl1*-Δ*dcl2* mutant libraries had very similar distribution profiles along (+) and (−) strands of the TMV genome, as the position of major and minor siRNA hotspots were parallel between these libraries (Figure 5), while their distribution profiles were also relatively similar to those in the TMV-infected *N. tabacum* libraries, particularly the Nt-2 library, but less similar to those in the Nt-1 library, which had much more abundant TMV siRNAs than the Nt-2 library (Figure 1E and Figure 5). Overall, these results showed that TMV siRNAs produced in *F. graminearum* have similar characteristics to those produced in plant hosts.

## 4. Discussion

The siRNAs and other classes of sRNAs play a central role in the RNA silencing pathway [1,5]. It is known that plants encode at least four DCL proteins [69]. In *Arabidopsis thaliana*, DCL4 and DCL2 are involved in antiviral defense against RNA viruses. DCL4 plays the major role in intracellular antiviral silencing and generates 21-nucleotide-long vsiRNAs, while DCL2 generates 22-nucleotide-long vsiRNAs and functions in intercellular (systemic) antiviral silencing. However, DCL2 can substitute for the antiviral functions of DCL4 when the latter is absent or repressed, although with lower efficiency [70,71,72,73,74,75]. DCL3 is involved in antiviral responses against DNA viruses and produces 24-nucleotide-long vsiRNAs [76,77]. Thus, RNA virus siRNAs in plants are predominantly 21 or 22 nucleotides in length, with peaks at 21 nucleotides. On the other hand, numerous studies have revealed that RNA mycovirus-derived sRNAs tend to have varied size profiles and broad size distributions among different viruses and fungal hosts. In *F. graminearum*, siRNAs derived from different mycoviruses also have broad size distributions, with a slight peak at 21 nucleotides [41,55]. Characterization of vsiRNA derived from taxonomically diverse mycoviruses infecting *Botrytis cinerea* showed that 20, 21, and 22 nucleotides are the predominant sizes, with the most abundant size class at 21 nucleotides, or 22 nucleotides for a certain mycovirus [53]. In *Aspergillus fumigatus*, the most abundant vsiRNA size class is 20, 21, or 20–23 nucleotides, depending on the virus [54]. Likewise, siRNAs of mitoviruses in *Fusarium circinatum* and *C. parasitica* have been found to have broad size distributions [46,58]. Together, these observations suggest that RNA genomes of various mycoviruses are processed differently by the fungal DCL to generate vsiRNAs.

Our results showed that the abundances of CMV and TMV siRNAs in the fungal hosts were much lower compared to those in the host plants (Figure 1B,E). This is likely due to the lower CMV and TMV accumulation in fungi than in plants as observed in our previous studies [63,66]. It is intriguing that the characteristics of CMV and TMV siRNAs generated in the fungal hosts are distinct from the siRNAs of mycoviruses, but instead similar to the characteristics of CMV and TMV siRNAs generated in plant hosts. This may suggest that fungal DCLs have adaptive capabilities in the manner of recognizing and processing various types of incoming viruses. Although the antiviral role of DCLs of some ascomycetous fungi has been elucidated, the signatures of vsiRNA products of a particular fungal DCL are still unclear. In *F. graminearum*, DCL1 and DCL2 function redundantly in antiviral defense against TMV infection [63]. The results of our analysis on TMV siRNAs in *F. graminearum* single Δ*dcl1* and Δ*dcl2* mutant strains could not reveal the specific sizes of the vsiRNA products of each DCL. The abundance of TMV siRNA strongly decreased in both Δ*dcl1* and Δ*dcl2* mutant strains, but here vsiRNAs still retained lengths of predominantly 21 and 22 nucleotides, and the same 5′-terminal nucleotide preferences as in the wild-type strain (Figure 4). Thus, in contrast with plants, in which each DCL generates siRNAs with a distinct size, single fungal DCL may produce heterogeneous sizes of siRNAs, although some size preferences may exist. Supporting this view, in *Mucor circinelloides*, DCL2 produces two different sizes (21 and 25 nucleotides) of antisense siRNAs that are associated with RNA silencing induced by the hairpin RNA-producing transgene [78]. The differential profiles of siRNAs between plant and fungal viruses may reflect substantial differences in replication processes between plant and fungal viruses; examples include differences in the replication site, type of membranes, and other cellular factors involved in viral replication. These differences may determine how DCLs gain access to and process virus-derived dsRNA substrates such as viral dsRNA replication intermediates and/or highly structured virus RNA genomes that are used for biogenesis of virus-derived siRNAs [79].

Although deletion of either the *dcl1* or *dcl2* gene markedly reduced TMV siRNA accumulation in *F. graminearum*, unexpectedly, deletion of both *dcl1* and *dcl2* genes restored TMV sRNA accumulation (Table 2 and Figure 1E). Moreover, TMV sRNAs accumulated in the double Δ*dcl1*-Δ*dcl2* mutant strain have similar size distributions, 5′-terminal nucleotide preferences, and distribution profiles along the viral genome to those accumulated in the wild-type strain (Figure 4 and Figure 5). These results showed that *F. graminearum* DCL1 and DCL2 are involved in the production of TMV siRNAs. However, in the absence of both DCL1 and DCL2, there is an activation of a hitherto DCL-independent pathway to produce virus-derived sRNAs that have similar signatures to DCL-dependent vsiRNAs. As deletion of both *dcl1* and *dcl2* genes highly elevated TMV accumulation levels in *F. graminearum* [63], these highly accumulated DCL-independent TMV-derived sRNAs appear to be unable to mediate RNA silencing antiviral defenses, although they have similar features to the bona fide vsiRNAs produced by the DCLs. Similarly, a high accumulation of DCL-independent vsiRNAs with no antiviral function was also observed in *dcl* mutant strains of *Cryphonectria parasitica* and *Colletotrichum higginsianum*, but these DCL-independent vsiRNAs have different characteristics to DCL-dependent vsiRNAs [40,80]. Together, these observations indicate that in fungi, an alternative DCL-independent pathway processes virus-derived dsRNA precursors when functional DCLs are absent. It is still unclear what kind of fungal RNA endonuclease can alternatively cleave the viral RNAs. Previously, the presence of endogenous DCL-independent sRNAs had been identified in other fungi. In the basal fungus *Mucor circinelloides*, RdRP-dependent dicer-independent sRNAs are generated by RNase III-like protein specifically found in basal fungi and involved in the specific degradation of cellular mRNAs [81]. In *N. crassa*, DCL-independent siRNAs were found to mediate DNA methylation [34,82]. Further studies into the mechanisms underlying the biosynthesis of these DCL-independent vsiRNAs in fungi, and their biological significance, would be of interest.

Mycoviruses are mostly transmitted horizontally through hyphal anastomosis and vertically through sexual or asexual spores [35], but current knowledge has established that during fungal colonization, a plant and fungus can exchange macromolecules, including protein effectors, siRNAs, and nucleic acid parasites such as viruses and viroids [16,63,64,66,83]. From a broader evolutionary perspective, the finding that RNA silencing in fungi is functional and effective in conferring antiviral responses against plant viruses and viroids may indicate that fungal antiviral RNA silencing has also evolved to cope with cross-kingdom infection by the native molecular parasites of plants. 

## Figures and Tables

**Figure 1 biology-11-01672-f001:**
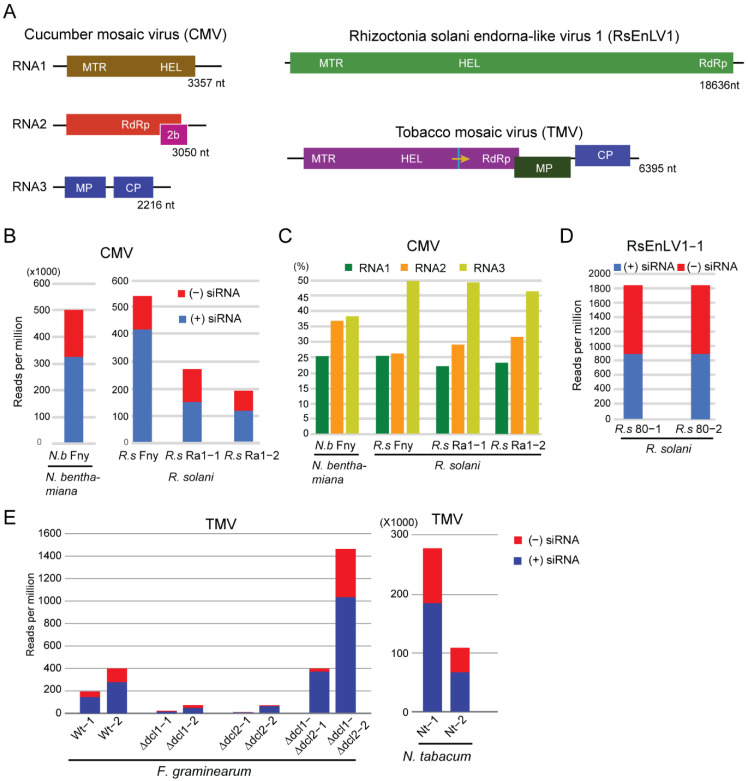
Abundances of mycovirus and plant virus siRNAs. (**A**) Schematic representation of the genome structure of cucumber mosaic virus (CMV), Rhizoctonia solani endorna-like virus (RsEnLV1), and tobacco mosaic virus (TMV) used in this study (not to scale). MTR, methyltransferase motif; HEL, helicase motif; RdRP, RNA-dependent RNA polymerase motif; MP, movement protein; CP, coat protein. (**B**) Normalized read number (read per million) of CMV siRNAs in *R. solani* and *N. benthamiana* sRNA libraries. “(−)” and “(+)” indicate siRNAs derived respectively from the complementary (negative) or positive viral genomic strands. (**C**) Proportion of siRNAs mapped to CMV RNA1, 2, and 3. (**D**) Normalized read number of RsEnLV1 siRNAs in *R. solani*. (**E**) Normalized read number of TMV siRNAs in *F. graminearum* and *N. tabacum* sRNA libraries.

**Figure 2 biology-11-01672-f002:**
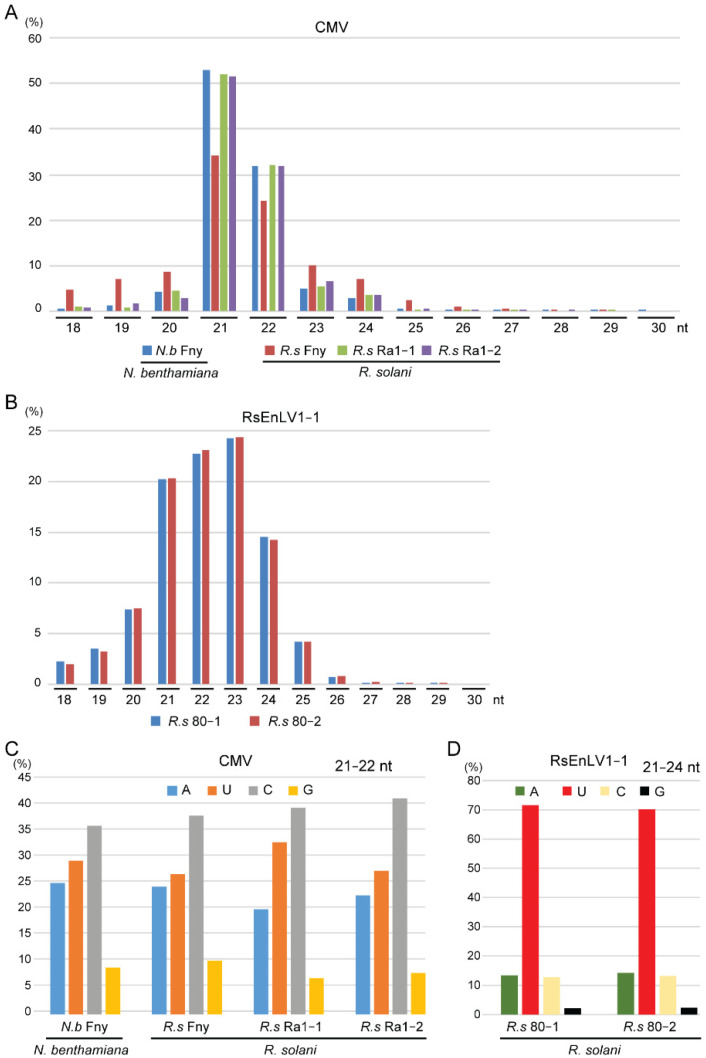
Characteristic of CMV and RsEnLV1 siRNAs. (**A**,**B**) Size distribution (percentage) of CMV siRNAs (**A**) and RsEnLV1 siRNAs (**B**). (**C**,**D**) Proportion of the 5′-terminal nucleotide of siRNA derived from CMV (**C**) and RsEnLV1 (**D**).

**Figure 3 biology-11-01672-f003:**
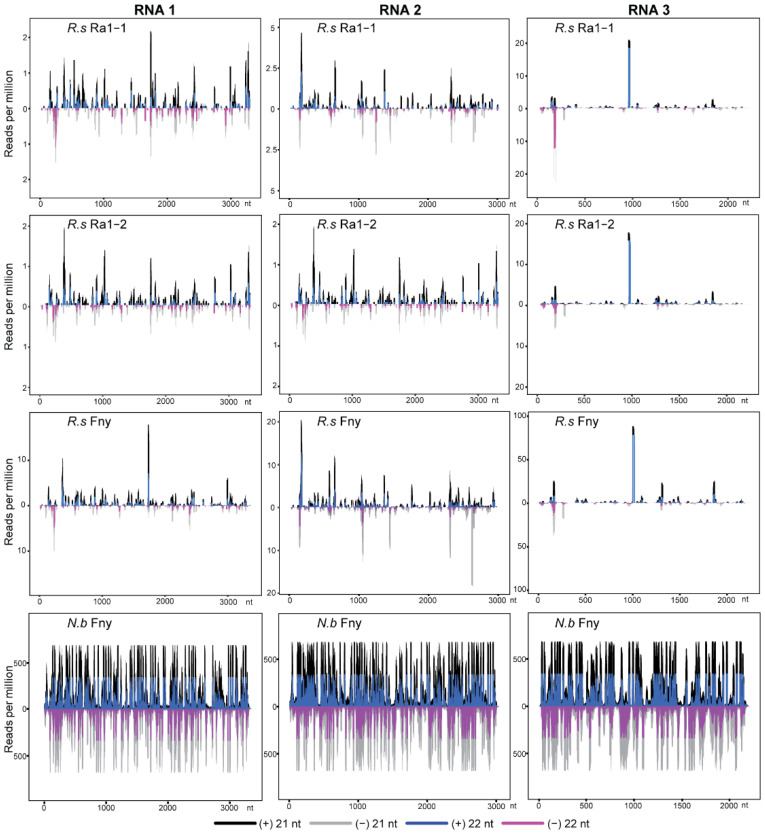
Distribution of CMV siRNAs (21 and 22 nt) along the viral genome in *R. solani* and *N. benthamiana* libraries. “(−)” and “(+)” indicate siRNAs derived respectively from the complementary (negative) or positive viral genomic strands.

**Figure 4 biology-11-01672-f004:**
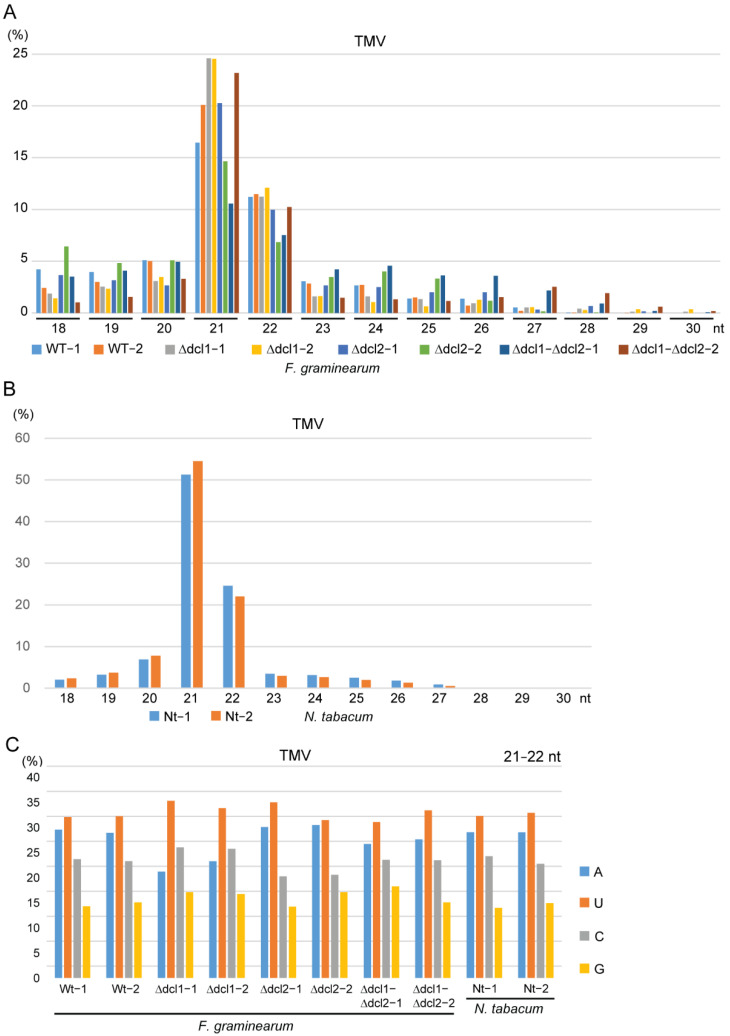
Characteristics of TMV siRNAs. (**A**,**B**) Size distribution (percentage) of TMV siRNAs in *F. graminearum* (**A**) and *N. tabacum* sRNA libraries. (**C**) Proportion of the 5′-terminal nucleotide of TMV siRNAs in *F. graminearum* and *N. tabacum* sRNA libraries.

**Figure 5 biology-11-01672-f005:**
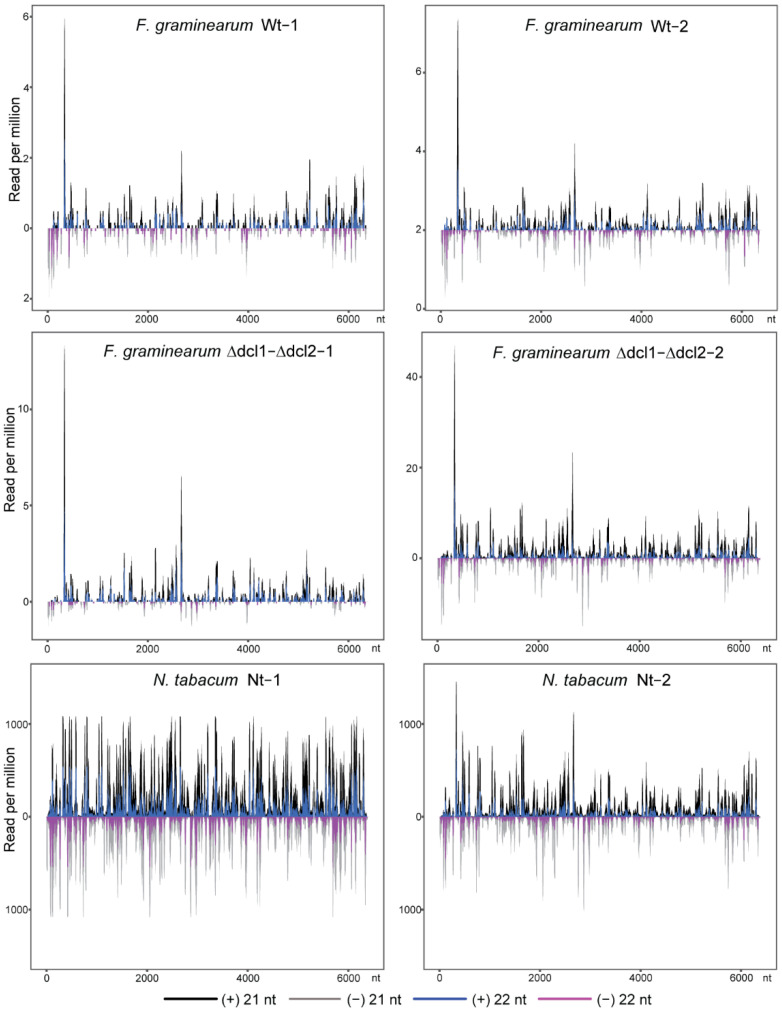
Distribution of TMV siRNAs (21 and 22 nt) along the viral genome in *F. graminearum* and *N. tabacum* sRNA libraries. “(−)” and “(+)” indicate siRNAs derived respectively from the complementary (negative) or positive viral genomic strands.

**Table 1 biology-11-01672-t001:** Read numbers of CMV and mycovirus siRNAs in *R. solani* and *N. benthamiana*.

Sample (Library)	*R. solani* Ra1-1	*R. solani* Ra1-2	*R. solani* Ra1/CMV-Cured-1	*R. solani* Ra1/CMV-Cured-2	*R. solani* /CMV Fny	*N. benth-* *amiana*/CMV Fny	*R. Solani* 80-1	*R. Solani* 80-2	*R. Solani* 80 /Virus-Cured-1	*R. Solani* 80 /Virus-Cured-2
Total sRNA	16,482,261	20,302,809	21,988,288	21,827,663	17,824,097	23,621,525	18,828,697	18,258,925	25,667,410	25,075,982
Virus	CMV Rs	CMV Rs	CMV Rs	CMV Rs	CMV Fny	CMV Fny	RsEnLV1	RsEnLV1	RsEnLV1	RsEnLV1
siRNAs	4455	3885	0	2	11,738	11,817,040	34,391	33,911	131	107
Percentage	0.03%	0.02%	0.00%	0.00%	0.07%	50.03%	0.18%	0.19%	0.00%	0.00%

**Table 2 biology-11-01672-t002:** Read numbers of TMV siRNAs in wild-type and *dcl* mutants of *F. graminearum* and *N. tabacum*.

Sample (Library)	*F. graminearum*										*N. tabacum*	
	Wt/Virus Free-1	Wt/Virus Free-2	Wt-1	Wt-2	Δ*dcl1*-1	Δ*dcl1*-2	Δ*dcl2*-1	Δ*dcl2*-2	Δ*dcl1*-Δ*dcl2*-1	Δ*dcl1*-Δ*dcl2*-2	Nt-1	Nt-2
Total sRNA	7,874,233	9,424,347	12,282,580	11,953,464	11,791,300	9,358,452	16,637,401	12,346,680	11,849,385	12,552,172	14,797,683	11,033,408
siRNAs	0	0	2386	4787	374	707	301	935	4792	18,604	4,043,468	1,184,759
Percentage	0.00%	0.00%	0.02%	0.04%	0.00%	0.01%	0.00%	0.01%	0.04%	0.15%	27.33%	10.74%

## Data Availability

Data are available from corresponding author upon reasonable request.

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
