# Peer review of "Similar Characteristics of siRNAs of Plant Viruses Which Replicate in Plant and Fungal Hosts"

_biology, 2022, doi:10.3390/biology11111672_

Round 1

Reviewer 1 Report

The article described vsiRNA profiles in two plant RNA virus-fungal host pathosystems: cucumber mosaic virus (CMV) infection in phytopathogenic fungus Rhizoctonia solani and tobacco mosaic virus (TMV) infection in phytopathogenic fungus Fusarium graminearum. The relative abundances and characteristics of siRNAs derived from CMV and TVMV were also addressed. Especially the evidence that siRNAs production is associated with DCL protein indicates the fungal and plants are sharing similar RNA silencing pathway.

   The manuscript is well written and meets the quality standard of journal. Some points remain to be revised:

(1)SiRNA sequencing was often used to identify the siRNA population in viral infected samples. Here, through small RNA sequencing, author analyzed the characteristics of CMV- and mycovirus-derived siRNAs in R. solani. SiRNAs distribution of the viral genome (CMV or TMV) in corresponding hosts was shown in Figure 3 and Figure5.

Especially, in figures, the siRNA reads mapping to RNA3 of CMV was obviously less than those siRNAs reads in RNA1 and RNA2. It’s interesting that the siRNA profile patter of RNA3 is quite conserved in R.s Ra1-1, R.s Ra1-2 and R.s Fny. However, due to the defects of deep sequencing, it’s possibly an artifact peaks. To confirm the siRNA accumulation, thus, it is suggested that author should quantify the relative siRNA accumulation level by Northern blot or siRNA qPCR. For example, the siRNAs with high reads, middle reads, few reads or 0 reads were selected to verify through specific siRNA probes or primers. Especially author should pay attention to that that conserved hot spot with highest number of siRNA reads in three samples.

(2) in Figure 5, the siRNA profile of TMV in F. graminearum Wt-1 and Wt-2 are quite similia, but different with F. graminearum Δdcl1dcl2-1. siRNA profile of TMV in F. graminearum Wt-2 is almost the same as F. graminearum double Δdcl1dcl2-2. Please check if it is a mistake. For easy comparison, the value of Y-axis in F. graminearum Wt-1 and Wt-2, F. graminearum Δdcl1dcl2-1 andΔdcl1dcl2-2 should be the same in maximum 6 or minimum -2. The similar modification should be done in Figure 3 as well.  

(3) In line 246-250, author described that “Their distribution profiles were also relatively similar to those in the TMV-infected N. tabacum libraries (Figure 5)”.  As shown in Figure 5, siRNA profile in N. tabacum Nt-1 is not similar to that of F. graminearum Wt-1 and Wt-2, F. graminearum Δdcl1dcl2-1 andΔdcl1dcl2-2. Please check it.

(4) in line 251, author concluded that “Overall, these results showed that TMV siRNAs produced in  F. graminearum have similar characteristics to those produced in plant hosts, and that F.  graminearum DCL1 and DCL2 are involved in the production of TMV siRNAs”. I would like to understand that F.  graminearum DCL1 and DCL2 might not be involved in siRNA production because TMV siRNAs are similarly processed in F.  graminearum Wt samples (with DCL1 and DCL2) and F.  graminearum double mutants samples(no DCL1 and DCL2). This is indicating that siRNA production of TMV is independent on DCL1 and DCL2. Thus, author’s the speculation that “However, in the absence of both DCL1 and DCL2, there is activation of a hitherto DCL-independent  pathway to produce virus-derived sRNAs that have similar signatures to DCL-dependent vsiRNAs.” is not solid with the limited evidence. Of course, It is not a problem to discuss in discussion part.  

(5)By the way, I assume that name of “F. graminearum Δdcl1dcl1-2” is a mistake. It should be “F. graminearumΔdcl1dcl2-1”. Please confirm it.

Author Response

Reviewer 1

- Thank you very much for your interest and positive comment on our work.

(1) SiRNA sequencing was often used to identify the siRNA population in viral infected samples. Here, through small RNA sequencing, author analyzed the characteristics of CMV- and mycovirus-derived siRNAs in R. solani. SiRNAs distribution of the viral genome (CMV or TMV) in corresponding hosts was shown in Figure 3 and Figure5.

Especially, in figures, the siRNA reads mapping to RNA3 of CMV was obviously less than those siRNAs reads in RNA1 and RNA2. It’s interesting that the siRNA profile patter of RNA3 is quite conserved in R.s Ra1-1, R.s Ra1-2 and R.s Fny. However, due to the defects of deep sequencing, it’s possibly an artifact peaks. To confirm the siRNA accumulation, thus, it is suggested that author should quantify the relative siRNA accumulation level by Northern blot or siRNA qPCR. For example, the siRNAs with high reads, middle reads, few reads or 0 reads were selected to verify through specific siRNA probes or primers. Especially author should pay attention to that that conserved hot spot with highest number of siRNA reads in three samples.

Answer: As presented in Figure 1C, the proportion of siRNA reads mapped to RNA3 is higher to those mapped to RNA1 or RNA2. It is a good suggestion to carry out further molecular detection to confirm the presence of abundant siRNAs corresponding to the hot spots, however, in the current study we focus to characterize virus siRNAs based on bioinformatic analyses and we may carry out such detections in the future study. Moreover, It is important to note that we used biologically replicate samples for most of the analyses. In our opinion, this strengthens our data.

(2) in Figure 5, the siRNA profile of TMV in F. graminearum Wt-1 and Wt-2 are quite similia, but different with F. graminearum Δdcl1-Δdcl2-1. siRNA profile of TMV in F. graminearum Wt-2 is almost the same as F. graminearum double Δdcl1-Δdcl2-2. Please check if it is a mistake. For easy comparison, the value of Y-axis in F. graminearum Wt-1 and Wt-2, F. graminearum Δdcl1-Δdcl2-1 andΔdcl1-Δdcl2-2 should be the same in maximum 6 or minimum -2. The similar modification should be done in Figure 3 as well. 

Answer: The plot profile of TMV siRNAs in F. graminearum Wt-2 looks almost the same as that in F. graminearum double Δdcl1-Δdcl2-2 but it is not exactly the same if we look it in detail, and we have confirmed that this is not a mistake. The value of Y-axis in the plots (graphs) of mapped siRNA reads were based on normalized number (reads per million) of each library, therefore, we can not make the same value for all plots as the abundances of siRNA reads are different among sRNA libraries.

(3) In line 246-250, author described that “Their distribution profiles were also relatively similar to those in the TMV-infected N. tabacum libraries (Figure 5)”.  As shown in Figure 5, siRNA profile in N. tabacum Nt-1 is not similar to that of F. graminearum Wt-1 and Wt-2, F. graminearum Δdcl1-Δdcl2-1 andΔdcl1-Δdcl2-2. Please check it.

Answer: We have modified the sentence for better description. Line 321-328

(4) in line 251, author concluded that “Overall, these results showed that TMV siRNAs produced in  F. graminearum have similar characteristics to those produced in plant hosts, and that F.  graminearum DCL1 and DCL2 are involved in the production of TMV siRNAs”. I would like to understand that F.  graminearum DCL1 and DCL2 might not be involved in siRNA production because TMV siRNAs are similarly processed in F.  graminearum Wt samples (with DCL1 and DCL2) and F.  graminearum double mutants samples (no DCL1 and DCL2). This is indicating that siRNA production of TMV is independent on DCL1 and DCL2. Thus, author’s the speculation that “However, in the absence of both DCL1 and DCL2, there is activation of a hitherto DCL-independent  pathway to produce virus-derived sRNAs that have similar signatures to DCL-dependent vsiRNAs.” is not solid with the limited evidence. Of course, It is not a problem to discuss in discussion part. 

Answer: We have moved these statements to discussion section. Line 404-412

(5) By the way, I assume that name of “F. graminearum Δdcl1-Δdcl1-2” is a mistake. It should be “F. graminearumΔdcl1-Δdcl2-1”. Please confirm it.

Answer: Thank you for pointing out this mistake. We have corrected the label in Figure 5

Reviewer 2 Report

Several research groups including the authors’ one have previously shown plant viruses to be able to be hosted in fungi. Indeed, some plant viruses and viroids are detected from field-collected fungal isolates. It has been an interesting research topic to investigate how two totally different cellular organisms protect themselves against single viruses. RNA silencing is the primary antiviral defense in plant and fungi. The manuscript no biology-2033454 by Pang et al. analyzed vsiRNA profiles for TMV and CMV in fungal and plant hosts. These two viruses are model plant viruses and has been utilized to greatly advance virology. Interestingly, the authors showed some differences and commonalities between vsiRNA production patterns in the two different kingdoms of hosts. Both CHV- and TMV-derived siRNAs were accumulated much less in the fungal hosts Rhizoctonia solani and Fusarium graminearum, respectively, than in the plant hosts Nicotiana benthamiana and Nicotiana tabacum. More peaks of vsiRNA production spanned along the genomic segments in plants than in fungi. Otherwise, the two host kingdoms showed similar patterns in their size distributions or classes, proportion of plus and minus senses, and 5’-terminal nucleotide preference. Moreover, the authors showed dcl-independent TMV-derived small RNA production in infected F. graminearum without any sign of antiviral effects (Bian et al., PNAS 2020), which exhibited similar profiles for the dcl-dependent siRNA production. The last finding warrants further investigation in the future.

This study is an extension of the authors’ previous studies (Andika et al., PNAS 2017; Wei et al., PNAS 2019; Bien et al., PNAS 2020; Cao et al., Viruses 2022) and contributes to a better understanding of how different host kingdoms employ antiviral RNA silencing. The manuscript is straightforward, well-written and smooth in most parts, with interesting new data. There follow some comments for further improvement:

Major points:

First, this reviewer wondered if the low abundance of vsiRNAs in fungal hosts reflected their virus titres. Second, relatively large proportions of CMV RNA3-derived siRNAs were noted (Fig. 1C), compared to RNA1- or RNA2-derived siRNAs. Third, there is a host spot of plus-strand siRNA observed on RNA3. Is this associated with subgenomic RNA production or the intergenic region of RNA3. These points deserve discussion in appropriate places.

Minor points:

Line 12. Spell “siRNA” out.

Line 16. This reviewer would suggest that “two” be replaced with “multiple,” as the authors are dealing with more than two pathosystems. The authors used two viruses and a total of four host organisms.

Line 43. Should replace “implicated” with “indicated.”

Line 47. Should be more specific, as some yeast species retain RNA silencing machineries. Or rephrase it.

Lines 70, 72, and possibly in other places. Should be “ascomycetous.”

Lines 81, 82. “Saccharomyces cerevisiae” rather than “yeast” needs to be used here.

Line 137. Spell “sRNA” out at its first appearance.

Line 148. Incorrect. Three-segmented.

Line 217. Needs a reference.

Line 278. Should be “broad peak.”

Line 300. Add “single” before “fungal.”

Line 312. Should be “sRNA” not “siRNA.”

Line 336. Should be “From” rather than “On.”

Line 339. Should be “cross-kingdom?”

Table 1. Decimal separators should be periods not commas in English.

Author Response

Reviewer 2

-Thank you very much for your interest and positive comment on our work.

Major points:

First, this reviewer wondered if the low abundance of vsiRNAs in fungal hosts reflected their virus titres.

Answer: We have added the discussion regarding this matter in discussion section. Line 372-375

Second, relatively large proportions of CMV RNA3-derived siRNAs were noted (Fig. 1C), compared to RNA1- or RNA2-derived siRNAs.

Answer: We have added the short discussion regarding this matter in result section. Line 237-239

Third, there is a host spot of plus-strand siRNA observed on RNA3. Is this associated with subgenomic RNA production or the intergenic region of RNA3. These points deserve discussion in appropriate places.

Answer: We have added the discussion regarding this matter in result section. Line 274-279

Minor points:

Line 12. Spell “siRNA” out.

Answer: The texts have been added.

Line 16. This reviewer would suggest that “two” be replaced with “multiple,” as the authors are dealing with more than two pathosystems. The authors used two viruses and a total of four host organisms.

Answer: In the text we have specified “two plant RNA virus-fungal host pathosystems”, thus we exclude plant hosts in this statement.

Line 43. Should replace “implicated” with “indicated.”

Answer: In our opinion, “implicated” is a proper word for this sentence.

Line 47. Should be more specific, as some yeast species retain RNA silencing machineries. Or rephrase it.

Answer: “yeast” has been removed.

Lines 70, 72, and possibly in other places. Should be “ascomycetous.”

Answer: The texts have been changed

Lines 81, 82. “Saccharomyces cerevisiae” rather than “yeast” needs to be used here.

Answer: The texts have been changed

Line 137. Spell “sRNA” out at its first appearance.

Answer: The texts have been added

Line 148. Incorrect. Three-segmented.

Answer: the texts have been changed

Line 217. Needs a reference.

Answer: A reference has been added

Line 278. Should be “broad peak.”

Answer: The text has been changed

Line 300. Add “single” before “fungal.”

Answer: texts have been added

Line 312. Should be “sRNA” not “siRNA.”

Answer: The text has been changed

Line 336. Should be “From” rather than “On.”

Answer: The text has been changed

Line 339. Should be “cross-kingdom?”

Answer: The text has been changed

Table 1. Decimal separators should be periods not commas in English.

Answer: The commas have been changed to periods

Reviewer 3 Report

Pang et al., characterized vsiRNA profiles through Hight-throughput seq from two plant RNA virus-fungal host pathosystems: cucumber mosaic virus (CMV) and tobacco mosaic virus (TMV)  infecting the phytopathogenic fungus Rhizoctonia solani and Fusarium graminearum, respectively. They have reported that fungal RNA silencing recognizes and processes plant RNA viruses similarly to plants. Interestingly, they found that CMV infection in R. solani was relatively associated with the accumulation of vsiRNAs. In addition, They showed that TMV siRNAs produced in F. graminearum have similar characteristics to those produced in plant hosts, and that F. graminearum DCL1 and DCL2 are involved in the production of TMV siRNAs. Overall the research sounds great and could be given a chance for publication; however, it needs some minor improvement. References such as "Sara Lopez-Gomollon, Roles of RNA silencing in viral and non-viral plant immunity and the crosstalk between disease resistance systems" could be referenced in this manuscript.

Abstract:

"Cucumber mosaic virus" and "Tobacco mosaic virus" are officially recognized by the International Committee on Taxonomy of Viruses; therefore, they could be italicized throughout the manuscript.

Line 12-13: Name the pathogenic fungi and viruses or mention some examples.

"Via" should be italicized

Line 19: replace "compared to" with "than"

Material and methods

Although most methods were referenced, it would be better to detail them. Furthermore, the authors could split the methods into different sub-sections.  

Lines 138-139: were used

Line 139: which corresponding figures?

The authors should provide the data or the accession number (although they said it is under process)

Results:

Line 145-152: As the authors already mentioned (https://doi.org/10.1073/pnas.1714916114), CMV characterization was already conducted previously. It wasn't conducted in this study. Therefore, the characterization in the result section is misplaced. As well as figure 1A.

Discussion:

Line 266-286: This paragraph sounds more like an introduction than a discussion.

Author Response

Reviewer 3

Overall the research sounds great and could be given a chance for publication; however, it needs some minor improvement. References such as "Sara Lopez-Gomollon, Roles of RNA silencing in viral and non-viral plant immunity and the crosstalk between disease resistance systems" could be referenced in this manuscript.

Answer: Thank you very much for your interest and positive comment on our work. The reference has been added to the manuscript.

Abstract:

"Cucumber mosaic virus" and "Tobacco mosaic virus" are officially recognized by the International Committee on Taxonomy of Viruses; therefore, they could be italicized throughout the manuscript.

Answer: According to ICTV, a virus name should never be italicized and should be written in lower case (https://ictv.global/faqs).

Line 12-13: Name the pathogenic fungi and viruses or mention some examples.

Answer: Information have been added.

"Via" should be italicized

Answer: The text has been italicized.

Line 19: replace "compared to" with "than"

Answer: The texts have been changed.

Material and methods

Although most methods were referenced, it would be better to detail them. Furthermore, the authors could split the methods into different sub-sections. 

Answer: As suggested, we have revised the method section.

Lines 138-139: were used

Answer: The text has been added.

Line 139: which corresponding figures?

Answer: The texts have been changed to “the plot of siRNA mapping” for clarity.

The authors should provide the data or the accession number (although they said it is under process)

Answer: Accession numbers have been added.

Results:

Line 145-152: As the authors already mentioned (https://doi.org/10.1073/pnas.1714916114), CMV characterization was already conducted previously. It wasn't conducted in this study. Therefore, the characterization in the result section is misplaced. As well as figure 1A.

Answer: In this paragraph (including Figure 1A), we briefly introduce the CMV strains that were used for analyses. We intentionally place this information in the result section to help the reader to follow the results easily.

Discussion:

Line 266-286: This paragraph sounds more like an introduction than a discussion.

Answer: In this paragraph we describe the general characteristics of siRNAs derived plant and fungal viruses revealed by various studies. In our opinion, to enable better interpretation and discussion regarding our results, this information is necessary to be presented in the beginning of discussion section.

Reviewer 4 Report

This study investigated vsiRNA profiles of two plant viruses in plant and fungal hosts, in which revealed that qualitative and quantitative characteristics of vsiRNAs. 

I felt the paper contained interesting findings and well organized. I have two comments as below. 

1. I wonder why accumulation levels of vsiRNAs from CMV and TMV in plant hosts are significantly higher than those in fungal hosts. 

2. I wonder why vsiRNA accumulation level of TMV was elevated in double dcl1 and 2 knock-out mutant of F. graminearum. I recommend the authors should check accumulation level of TMV-vsiRNAs in the mutants by another experiment (e.g. northern blotting) rather than bioinfomatics analysis.  

Author Response

Reviewer 4

This study investigated vsiRNA profiles of two plant viruses in plant and fungal hosts, in which revealed that qualitative and quantitative characteristics of vsiRNAs.

I felt the paper contained interesting findings and well organized. I have two comments as below.

Thank you very much for your interest and positive comment on our work

  1. I wonder why accumulation levels of vsiRNAs from CMV and TMV in plant hosts are significantly higher than those in fungal hosts.

Answer: We suggest that this is due to the higher level of CMV and TMV accumulation in plant hosts than in fungal hosts. We have added this statement in discussion section. Line 372-375

  1. I wonder why vsiRNA accumulation level of TMV was elevated in double dcl1 and 2 knock-out mutant of F. graminearum.

Answer: This is likely due to highly elevated TMV accumulation levels in F. graminearum double dcl1 and dcl2 mutant. The discussion regarding this matter is presented in discussion section. Line 412-415

I recommend the authors should check accumulation level of TMV-vsiRNAs in the mutants by another experiment (e.g. northern blotting) rather than bioinfomatics analysis. 

Answer: It is a good suggestion to carry out further analyze to verify the accumulation of TMV siRNAs in the fungal mutant, however, in the current study we focus to characterize virus siRNAs based on bioinformatic analyses and we may carry out such analyses in the future study. It is important to note that similar results were obtained from two biologically replicate samples for TMV siRNA analysis. In our opinion, this strengthens our conclusion.